# A Qualitative Investigation of the Psychosocial Impact of Chronic Low Back Pain in Ghana

Josephine Ahenkorah Ampiah ![ORCID] ,[1,2] Fiona Moffatt,[2] Claire Diver,[2] Paapa Kwesi Ampiah[2]

[1]Division of Physiotherapy, Chiropractic and Sports Rehabilitation, London South Bank University, London, UK
[2]Department of Physiotherapy and Rehabilitation Sciences, University of Nottingham, Nottingham, UK

**Correspondence to**
Dr Josephine Ahenkorah Ampiah; ampiahj2@lsbu.ac.uk

## ABSTRACT

**Introduction** Chronic low back pain (CLBP) is a global health concern associated with multidimensional/biopsychosocial levels of affectation in developed countries, with holistic management requiring consideration of these factors. There has been minimal research exploring the psychosocial impact of CLBP, and the factors influencing it, in African contexts, with none in Ghana.

**Objectives** To explore the psychosocial impact of CLBP among patients with CLBP in Ghana.

**Design** Qualitative study using individual semistructured face-to-face interviews, underpinned by Straussian grounded theory principles and critical realist philosophy.

**Participants** Thirty patients with CLBP attending physiotherapy at two hospitals in Ghana.

**Results** Five categories: loss of self and roles, emotional distress, fear, stigmatisation and marginalisation, financial burden, and social support and three mechanisms: acquired biomedical/mechanical beliefs from healthcare professionals (HCPs), sociocultural beliefs and the socioeconomic impact of CLBP were derived.

**Conclusion** CLBP adversely affects multidimensional/biopsychosocial aspects of individuals experiencing CLBP in Ghana. This delineates the need for a biopsychosocial approach to care. There is the need for HCPs in Ghana to reassess current CLBP management strategies to address the influence of adverse HCPs biomedical inclinations on patients' psychosocial consequences. Population-based education strategies and consideration of formal support systems for persons with disabling CLBP may also be beneficial.

## STRENGTHS AND LIMITATIONS OF THIS STUDY

⇒ This study, using rigorous qualitative methods, demonstrates that similar adverse psychosocial impacts are present among patients experiencing chronic low back pain (CLBP) in developed countries and an African country (Ghana), although some disparities exist.

⇒ The use of purposive, theoretical and maximum variation sampling across two different geographical regions of Ghana strengthens the reliability of the findings.

⇒ An inductive and qualitative approach facilitated development of categories/findings that represented patients' own voices and provided in-depth explanations.

⇒ The findings may not be applicable to Ghanaians with CLBP who have not accessed the healthcare system or other African countries, due to differing contextual factors that may exist in different countries and situations.

⇒ Conducting member-checking through returning of findings to participants could have improved the confirmability and thus the validity of this study.

## INTRODUCTION

Low back pain (LBP) is the leading cause of disability globally.[1 2] The prevalence of LBP has risen significantly in the last three decades: a 54% increase from 1990 to 2015.[3] The prevalence of LBP is projected to continue to rise, especially in low-income and middle-income countries.[1] The estimated point, annual, lifetime prevalence of LBP in Africa is 39%, 57% and 47%, respectively.[4] According to Kahere *et al*,[5] the prevalence of chronic LBP (CLBP) in sub-Saharan Africa ranged from: 18% to 28% among the general population, 30% to 56% among workers and 22% to 59% among patients with LBP. There are limited high-quality population-based studies assessing the prevalence of LBP in Ghana. Furthermore, only few studies have assessed the prevalence of LBP among workers in Ghana.[6 7] Nonetheless, these few studies suggest that the burden of LBP in Ghana is substantial. CLBP has also been reported as a common condition for which patients attend physiotherapy in Ghana.[8] A population-based study on the prevalence of LBP among persons aged 50 and over in five countries (Ghana, Mexico, India, Russia, South-Africa and China), found an LBP prevalence of 41% among Ghanaian adults: the second highest prevalence recorded across the five countries.[9] The prevalence of LBP among Ghanaian taxi drivers and miners was found to be 34% and 67%, respectively.[6 7]

Additionally, the economic burden of LBP is substantial, mainly resulting from productivity losses and healthcare costs.[10 11]

The risk factors, causes and impact of CLBP are multifaceted: comprising biological, psychological and social domains.[12] The adverse effects of CLBP at varied levels (individual, family, societal/national) have been well documented in developed countries.[13–15] Therefore, clinical practice guidelines for the management of CLBP in developed countries suggest the use of the biopsychosocial model for the assessment and management of CLBP.[16 17] This fosters holistic understanding of the impact of CLBP and highlights important indicators to be considered in the management of CLBP.[12] Conversely, a qualitative study[18] and review[19] on management practices for LBP in Ghana and African countries suggest the predominance of biomedical and passive strategies (eg, electrotherapy, massage, prolonged pain medication use), and an absence of biopsychosocial strategies. However, exercises are commonly prescribed by Ghanaian physiotherapists.[8] Late health seeking and the use of herbal medication for CLBP in Ghana have also been reported by Ghanaian patients experiencing CLBP.[18] CLBP is a long-term and complex condition that may persist beyond tissue healing.[20] Therefore, reliance on biomedical and passive strategies may be problematic. The psychosocial impact as experienced by Ghanaian patients, considering reliance on management strategies which are inconsistent with the current evidence for CLBP management is unknown.

The role of psychosocial factors on LBP/CLBP outcomes may be conceptualised broadly under three domains: prognostic abilities, treatment effect (moderators or modifiers) and treatment mediators.[21] The prognostic abilities of psychosocial factors are evident in their abilities to affect/predict CLBP outcomes irrespective of the therapy administered.[22] The moderating role of psychosocial factors is evident through studies that show the relationship between baseline psychosocial factors, interventions and the associated outcomes.[23] Lastly, psychosocial factors as mediators of CLBP depicts how psychosocial factors play an intermediary role between intervention and outcome.[21] Moreover, the potential for CLBP to generate adverse psychosocial impacts has been discussed in Western contexts.[13] This depicts the pluralistic positioning of psychosocial factors.

The psychosocial impact of CLBP has been widely investigated in developed countries through qualitative studies exploring the lived experiences of individuals with CLBP.[24–26] This has resulted in four meta-syntheses exploring the lived experiences and/or psychosocial impact of CLBP.[13–15 27] These studies report an adverse impact of CLBP on all aspects of patients' lives: self, relationships, work, domestic tasks and social life. Adverse effects of CLBP such as feeling of loss of self, stigmatisation, marginalisation, shame, guilt and financial difficulties were reported. However, social support was acknowledged as having a positive impact on the experience of CLBP. However, studies on CLBP conducted in African countries mostly report outcomes related to pain, disability and quality of life, with assessment/exploration of psychological aspects largely absent. Only three studies have assessed psychological risk factors of CLBP in African countries: Nigeria,[28] Cameroon and[29] South-Africa.[30] Social aspects of CLBP that have been widely researched in African contexts pertain to the biomechanical factors precipitating CLBP and sociodemographic/economic factors.[6 31–33] To date, only two studies have previously explored the psychosocial impact of CLBP in African countries (rural Nigeria)[34] and Ethiopia.[35] This highlights the limited understanding of how CLBP impacts the psychosocial well-being of patients in African contexts. Both studies reported significant adverse effects on patient's psychosocial well-being, for example, loss of livelihoods and relationships. The experience and impact of pain is subjective and is affected by cognitive (eg, previous experience of pain), sociocultural (eg, cultural beliefs/roles), economic and structural factors (eg, accessibility to healthcare), which vary in different African, and cultural contexts (eg, rural–urban divide and different tribes).[18 36] In Ghana, the impact of CLBP and biomedically oriented management approaches on patients' social and psychological well-being are unknown. The drivers of these psychosocial consequences are also unknown. This may limit understanding of patients' health needs and the illness experience of patients experiencing CLBP in Ghana. Moreover, application of holistic/patient-centred management strategies for CLBP is dependent on in-depth understanding of patients' biopsychosocial needs.[20] Additionally, this research would provide information that could be relied on by healthcare providers and systems to help direct relevant healthcare services and support for patients' with CLBP in Ghana. This research aimed to explore the psychosocial impact of CLBP and its management among patients experiencing CLBP in Ghana and understand the factors that drive reported psychosocial consequences of CLBP among patients in Ghana.

## METHODS

This study was part of a broader qualitative research that sought to explore different aspects related to the experience and management of CLBP among patients with CLBP in Ghana. The overarching research investigated aspects around two major domains: the CLBP beliefs and management practices, and the impact of CLBP and its management among Ghanaian patients experiencing CLBP. The methods used in this study have been previously published in an article explicating CLBP beliefs and management practices among Ghanaian patients experiencing CLBP.[18] A brief description of the methods used is presented below.

### Study design

Qualitative research design using in-depth individual semistructured face-to-face interviews.

## Methodology

Grounded theory (GT) methods as stipulated by Strauss and Corbin,[37] and a critical realist philosophy[38] underpinned this research, due to limited information around this area in the research context and the aim of deriving underlying mechanisms. Therefore, this research sought to highlight patients' explanations/stories regarding the impact of CLBP on their lives and the mechanisms facilitating the varied impacts discussed by patients. No initial theoretical framework was used, as suggested by Straussian GT principles.[37] Induction, deduction and abduction were used to derive categories, concepts and causal mechanisms/structures.[39–41] Discussions around the generation of a theory is beyond the scope of this paper.

## Study setting

The study was conducted in two hospitals (S1 and S2) serving Ghanaian patients located in different geographical regions (Southern belt and Middle/Northern belt) of Ghana; to enhance the breadth of patients' narratives and analysis of the contextual factors underlying the psychosocial impact of CLBP among Ghanaian patients.

## Sample

Purposive and maximum variation sampling were employed to enhance the depth/breadth of the narratives/study findings.[42 43] This study included patients across a broad range of ages, occupation and literacy. Theoretical sampling (as stipulated within GT) was also used to enhance in-depth understanding of emerging categories and verify emerging dimensions and relationships.[39] After purposively sampling 10 patients, theoretical sampling was initiated.

The inclusion criteria for this study were as follows: adult male and female participants (>18 years), presenting with LBP lasting more than 3 months[44] who were receiving physiotherapy at either study sites. The exclusion criteria were as follows: pregnant women and individuals diagnosed with specific or serious causes of CLBP (ie, trauma, infection, previous surgery, inflammatory causes or malignancy).[45 46]

## Recruitment

Two gatekeepers (physiotherapists) working at both study sites facilitated access to, and recruitment of participants. Participants were recruited from the outpatient physiotherapy departments of the two study sites. Recruitment spanned a period of 7 months (November 2018 to June 2019). The gatekeepers distributed information sheets to eligible participants or explained the content of the information sheets to participants who were either non-English speakers or illiterates. Follow-up reminders were carried out by the gatekeepers when patients attended physiotherapy. The first author made contact (in-person/ through telephone) with interested participants and arranged interview dates/times. A consent form and patient data capturing sheet were administered by the first author on the interview day.

## Patient/public involvement

Two patients with CLBP participated in a patient involvement session that aimed to introduce the research to patients and draw on their opinions and preferences. This facilitated planning of data collection (interview guides and conducting interviews). Two pilot interviews were conducted to assess appropriateness/suitability of the interview venue, content and structure of the interview. The pilot interviews facilitated improving the breadth of the prompt questions (eg, further break down of different aspects of life relevant to participants and probing how CLBP has affected these areas).

## Data collection

Data were collected by the first author, who situated herself as an insider (a Ghanaian physiotherapist with 10 years of physiotherapy clinical, teaching and research experience) and an outsider (someone who had never experienced LBP).[47] The interviews were conducted in a private room at both study sites, audiorecorded and lasted between 30 and 50 min. The interviews were conducted in English or Twi (for non-English-speaking participants). Data were collected until data saturation, that is, until no new information emerged, and all emerging dimensions had been fully explored.[47]

An interview guide derived from the research objectives and previous research[24 34 48] was used to guide the interviews. Two broad and open-ended questions and several prompts were used to elucidate the impact of CLBP and its management on patients' lives and the mechanisms at play: (1) What is your experience with low back pain? (Prompts: when and how did it start? how has it affected your daily activities or life (work, home/family and social activities)? (2) How have you managed your back pain since it started? (Prompts: How do you cope with the condition? What do you do to feel better? What makes you feel better? Where did you gain information regarding some of the coping strategies you are using? Do you patronise other alternative therapies aside physiotherapy or seeing a medical doctor?). The interview guide was translated to the local language (Twi) and back translated to English, and then the original version and back-translated version compared with ensure that the meaning was retained. To enhance theoretical sensitivity,[40] as data collection proceeded, other prompts were added to the interview guide (eg, how has CLBP affected your relationships?). Reflexive notes were taken throughout the research.[40]

## Data analysis

The processes for data analysis have been discussed in detail in a previously published article.[18] Data analysis was carried out using open and axial coding, induction, deduction, abduction and constant comparison of data.[37 39] All the interviews were transcribed verbatim by the first author. Out of 13 Twi transcripts, 5 were randomly

selected for back translation using the same process as the interview guide. Data were managed and stored by using NVivo V.12. Microsoft Word and traditional methods such as multiple photocopies, coloured pens and sticky notes were used to identify codes and relationships.[49]

Open coding and axial coding were carried out through line-by-line coding of each transcript; then descriptive and interpretative codes were inductively assigned and relationships between the codes established. Axial coding mainly involved identifying relationships between the codes, and the mechanisms underlying the codes being generated.[39] Identifying mechanisms were facilitated using the coding paradigm (conditions, actions, interactions, consequences).[38 50] Similar codes were grouped to form concepts; and concepts were grouped to form categories (eg, online supplemental file 1). A category thus contained concepts and their underlying mechanisms Interviews, codes, concepts, mechanisms and categories were constantly compared throughout analysis.[51] The rigour of the study was enhanced through the following processes: extensive engagement with the research contexts, use of maximum variation sampling, geographically different study settings, reflexivity, research team comprising expert qualitative researchers, all members of the research team reading all derived codes in the context of the raw data and all derived codes agreed on by the research team, consisting of expert qualitative researchers: FM and CD.

## RESULTS

Thirty patients from both study sites were involved in the study: S1-16 and S2-14 participants. They comprised 10 males and 20 females aged between 27 and 87 (mean±SD; 51.2±13.1). They were current and previous office workers (6), seamstresses (5), hospital workers (8), market women/traders (5), farmers (2), businessman (1), driver (1), planner (1), teacher (1), police officer (1), orderly (1) and journalist (1). The duration between LBP onset and attending hospital ranged from within the first year of LBP onset to 15 years; the time to then being referred for physiotherapy was 2 months to 25 years (table 1).

Five categories emerged from participants' narratives: (1) loss of self and roles, (2) emotional distress, (3) fear of the future and toxicity, (4) stigmatisation and marginalisation, (5) social support and financial burden. Three mechanisms underlying the psychosocial impact of CLBP were: the influence of healthcare professionals (HCPs') biomedical/mechanical and fear avoidance beliefs (FABs), sociocultural beliefs and socioeconomic impact. Ten concepts/subcategories were also derived (table 2). The HCPs that patients mainly referred to were doctors and physiotherapists. Participants expressed the feeling of loss of self, gendered roles, domestic roles and their livelihoods, feeling stigmatised and marginalised, distress and fear. They also described the financial implications associated with living with CLBP. Family/friends support was the main form of social support discussed by participants.

**Table 1** Sociodemographic characteristics of participants

|  | Frequency | % |
|---|---|---|
| **Age (years)** | | |
| 20–29 | 1 | 3.3 |
| 30–39 | 5 | 16.7 |
| 40–49 | 8 | 26.7 |
| 50–59 | 7 | 23.3 |
| 60–69 | 8 | 26.7 |
| 70–89 | 1 | 3.3 |
| **Sex** | | |
| Male | 10 | 33.3 |
| Female | 20 | 66.7 |
| **Previous/current occupation** | | |
| Office workers | 6 | 20.0 |
| Seamstresses | 5 | 16.7 |
| Hospital workers | 8 | 26.7 |
| Market women/traders/ businessmen | 5 | 16.7 |
| Farmers | 2 | 6.7 |
| Driver | 1 | 3.3 |
| Teacher | 1 | 3.3 |
| Police officer | 1 | 3.3 |
| Journalist | 1 | 3.3 |
| **Literacy** | | |
| Illiterate | 13 | 43.3 |
| Literate in English and/or Twi | 17 | 56.7 |
| **Duration between LBP onset and first medical visit** | | |
| <1 year | 10 | 33.3 |
| Between 1 year and 5 years | 13 | 43.3 |
| 5–10 years | 3 | 10.0 |
| >10 years | 4 | 13.3 |
| **Duration between first medical visit regarding LBP and being referred to physiotherapy** | | |
| <1 year | 11 | 36.7 |
| Between 1 year and 5 years | 11 | 36.7 |
| 5–10 years | 5 | 16.7 |
| >10 years | 3 | 10.0 |

LBP, low back pain.

It was reported as an indispensable source of psychosocial support throughout their journeys. However, participants' accounts demonstrated tendencies for family support to facilitate maladaptive behaviours.

### Category 1: Loss of Self and Roles
All the participants described some sort of loss resulting from CLBP. The different aspects of loss described by participants are reported throughout this section. Participants suggested that CLBP caused unpleasant experience of fluctuating pain, stiffness, altered posture and

**Table 2** Summary of categories, concepts and mechanisms

| Categories | Loss of self and roles | Fear | Emotional distress | Stigmatisation/ marginalisation | Social support |
|---|---|---|---|---|---|
| Concepts | Loss of self<br>▶ Unpleasant experience<br>▶ Loss of Spontaneity<br>▶ New self<br>Loss of roles<br>▶ Domestic roles<br>▶ Work roles/ livelihoods<br>▶ Gendered/cultural roles<br>▶ Hobbies | Fear of the future<br>▶ Disability<br>▶ Death<br>Fear of toxicity/side effects<br>▶ Orthodox medication use | Feelings of unhappiness/anger frustration/suicidal Ideations | Stigmatisation<br>Marginalisation | Financial burden<br>Family and friends support |
| Mechanisms | Patients' and HCPs' biomedical/ mechanical beliefs<br>Sociocultural beliefs<br>Socioeconomic impact | Patients' and HCPs' biomedical/ mechanical beliefs<br>Sociocultural beliefs | Patients' and HCPs' biomedical/ mechanical beliefs<br>Sociocultural beliefs<br>Socioeconomic impact | Patients' and HCPs' biomedical/ mechanical beliefs<br>Sociocultural beliefs<br>Socioeconomic impact | Patients and HCPs' biomedical/ mechanical beliefs<br>Sociocultural beliefs<br>Socioeconomic impact |

HCPs, healthcare professional.

disruption in movement and sleep; this resulted in a loss of their 'usual physical selves'. The 55-year-old shopkeeper felt CLBP had negatively impacted every aspect of her life.

> …it has changed every aspect. Pain, It has made me weak. Physically, even spiritually. Now I'm not like I previously was. I could wake up and work for long hours. I was very strong. Now…when I do something little, I get tired (P2S2).

Most participants reported a loss of spontaneity with performance of everyday activities. This was normally related to acquired biomechanical beliefs from HCPs (around posture and the need to protect the back), resulting in hypervigilance.

> Physically I manage because if I sleep and then I'm waking up, I need to get up strategically… Because if I bend down like this and I have to wake up, I need to take time gradually. So physically you see me but I'm not physically strong (P8S1).

The loss of their 'usual selves' resulted in psychological consequences. According to the participants, CLBP became a focal point of their thinking framework. They described regular periods of thinking about the pain, changes and difficulties that CLBP has brought into their lives and what the future with CLBP would entail. These thoughts were the main concerns associated with experience of a 'new self'.

> Because sometimes you look at what you could do and now you can't do them anymore… I was helpless… I cast my mind back and say why all these? Why is this happening to me? …Nothing is working for me

now. So psychologically, emotionally, I am tortured in a way (P7S1).

> It restricts your everything. The mind doesn't think properly. It doesn't even give you that atmosphere for you to work properly. Thinking about the pain… it's like anybody you see is an enemy. It becomes sensational. You don't even feel yourself (P1S2).

Participants recalled how CLBP resulted in the loss of roles which amplified the feeling of loss of identity. Participants felt that their roles as spouses, workers, parents and members of society constituted components of their identity. Therefore, alterations to or loss of any of the aforementioned roles (eg, working or spousal roles) affected their identities.

> …I love my job. I feel incapacitated because I cannot go and do what I love. I love to do stories. I am just there. I have been at home for one year and it affects everything about your life (P7S1).

All the participants expressed how CLBP had either caused loss of their livelihoods permanently or temporarily.

> Twelve years I worked in the bank. I never went for leave… and here I am. Not that I have resigned. They just deleted me from their books like that. When I started falling sick even financially, they never even supported. Sometimes I beg from my friends, my mates…and my extended family too because I was the breadwinner (P13S1).

A gendered perspective to loss was present in participants' narratives, with the inability to perform gendered or culturally prescribed domestic roles such as cooking

or sweeping reported by all the women. In very few cases, male participants mentioned loss of the ability to help with domestic activities, while emphasising they just played a complimentary role. Loss of the ability to perform sporting/gym activities was reported by only males (three), suggesting that differing gendered roles were affected by CLBP among men and women.

> It has affected the joy in the house. Some of the work in the house that you as an elderly woman should do, you give it to the children to do…, you cannot perform your duties (P4S2).

The influence of HCPs on participants' feeling of loss of self and roles was indicated by most of the participants. HCPs' advice on activity avoidance reportedly facilitated participants' loss of livelihoods and performance of domestic activities. Also, HCPs' prescription of multiple hospital visits (suggestive of a biomedical approach to care) influenced loss of participants' livelihoods.

> I was so devastated. Then I sat down and asked myself, is that how I'm going to be for the rest of my life? I mean they (HCPs) telling you not to do this, not to do that, don't drive…not having a normal life? (P14S2).

> I go to work once a week. Because you must go to the hospital and going to and fro the hospital you can't even get time to go to work (P4S1).

Most of the married participants in this study reported an influence of CLBP on their conjugal relationships, and this sometimes placed a strain on the marriage. These changes to participants' conjugal relationships were facilitated by FABs and biomedical beliefs derived from HCPs.

> …and especially too I have a wife (hisses) I normally I can't go to her because I get some pains. I have the psychological effect in the mind that when I do, I may get pains. So, I decided not to do it at all. I got to know from doctors that the disc is worn out so in fact ever since I've been cautious not to do the dos and don'ts (P6S1).

Participants' accounts described above highlight the influence of HCPs and the sociocultural environment (family, society and culture) on the CLBP experience and vice versa.

## Category 2: Emotional Distress

Most participants often recounted how their CLBP experiences involved periods of sadness and frustration, leading to distress. This was often expressed as being tired of the condition, not being happy or through anger. Distress was predominantly expressed in different ways among men and women in this study. Female participants expressed emotional distress by recounting periods of crying, and two males also recalled crying. However, only male participants (four) reported that they expressed their distress through anger directed at themselves and others.

> It has really worried me madam (hisses), because when it started, anytime I remember I cry (Cries) (P2S2).

> The least thing I become so angry at myself, even when you're giving me treatment, I see the treatment is not solving it quickly. I feel that aah! Sometimes I just have to go and relax. I would leave the scene, especially when my wife is with me (P13S1).

A former banker experienced a period of severe pain and movement restriction, coupled with the loss of his job without any entitlements. He reported that these resulted in extreme form of distress that led to suicidal thoughts. Other participants expressed their frustration with the persistence of the pain and the challenges associated with CLBP.

> One is the pain, two: my job… I used to become so silent just to endure the pain…It got to a point, I nearly caused suicide (P13S1).

Among the participants, this feeling of distress was often influenced by personal factors (the feeling of loss, severe pain, uncertainty about the future) and HCPs advice (radiological findings, FABs, hospital visits), and this was equally expressed in both male and female participants. Uncertainty about the future appeared to undermine renegotiation of a future identity.

> After doing the MRI and the doctor starts explaining the defects, I was so devastated. They were talking about sex, driving and all those things so I was asking myself is that how I'm going to be for the rest of my life? And I started crying. It's not just easy … having pains just for the rest of your life (P14S2).

## Category 3: Fear of the Future and Toxicity/Side-effects

The participants expressed fear of the future consequences of CLBP (mostly related to fear of disability). Some participants feared the spine would deteriorate further and cause disability. Hence the fear of disability was influenced by participants' biomedical orientation. In addition, some of the participants expressed the dislike for use of assistive devices (eg, walking sticks) because it facilitated being perceived as old or handicapped by oneself and others. This appeared to be facilitated by participants' personal beliefs.

> Sometimes I fear that I'm not that old, that if I start having this now then I don't know how it would be in a few years' time (P15S1).

One participant narrated a previous fear of death, as a result of excruciating pain. This fear of death had however been allayed by interactions with HCPs, according to this participant.

> Initially, I thought it would end my life, till I could see no, with the advice, with the physiotherapy, medications… (P13S1).

Another aspect of fear expressed by participants was the fear of toxicity/side effects related to the prolonged use of orthodox medication. Some participants reported that they were experiencing side effects of analgesics.

Formally, I was taking diclofenac. Now I have this epigastric pain, so diclofenac is not good for me (P9S1).

I started with pregacid, flotac, diclofenac and then it proceeded to morphine, codeine. I'm still taking it. I've taken a lot but for now three. Today he (doctor) added one to it so three. But I don't normally take it unless I feel the pain because I don't want to be addicted to it (P9S2).

Interestingly, participants did not express this fear of toxicity/side effects in relation to herbal medication. They believed herbal medication was from natural sources (eg, plants) hence had the potential to be less harmful, although they sometimes questioned the efficacy of herbal medication. Beliefs around herbal medication appeared to be underpinned by sociocultural beliefs. The local name for herbal medicine, used by the participants, when directly translated to English reads, 'African medicine'. Other participants also referred to it as local medicine.

…Orthodox medicine or whatever, all these drugs we are taking into our system, I didn't want the whole idea, so I decided that, though with the efficacy of the herbal drugs, the local medicine, it's another question of its own but then I believe that they are natural herbs (P7S1).

Generally, fear was influenced by derived biomedical/biomechanical perspectives and sociocultural beliefs as reported in participants' narratives above.

### Category 4: Stigmatisation and Marginalisation
Some participants' accounts depicted different aspects that caused feelings of being stigmatised or marginalised. This feeling was experienced at the family, work and societal levels. At the family level, two female participants felt misunderstood by their spouses. The source of this misunderstanding was the use of avoidance of activities as coping mechanisms.

Yes, it will translate to the husband. In the night you are not sleeping. So even when the man is coming, sex-wise it affects. You are not feeling comfortable with yourself. Aha and it is like nowadays you have changed the man doesn't understand the pain that is in you. But you are feeling it (P4S2).

At work, some participants felt marginalised. They felt that CLBP either served to make them an easy target for internal transfers or termination of appointment.

You see sometimes too nursing they don't even care. Comparing you to every other person. I always say I didn't know why they took me to Fevers unit… when

I started picking the excuse duty (sick-leave) left right then she (supervisor) started complaining… (P6S2).

At the societal level, a few participants reported that people felt CLBP was not reason enough to stop working, and so were tagged as lazy for not working. This portrays the sociocultural belief that CLBP may not be a serious illness. Another participant suggested that a lack of understanding of biomedical explanations for CLBP, offered the opportunity for others to construe the cause of LBP as spiritual, and this facilitated stigma.

The sewing for instance when I stopped, people felt I was lazy that's why I've stopped. They think you're lazy that's why you've stopped work (P5S1).

I called my sister-in-law that this my back has been hurting for long…I have even gone for an X-ray at Dr. B's place, and he said that it's the backbones that have widened up. So, she said can it happen that way? …. Later, she was trying to convince my husband that it is spiritual so my husband should leave me, because she hasn't seen a waist that has widened spaces (P3S1).

### Category 5: Social Support and Financial Burden
Social support may be in the form of employment support, government support or support received from relevant others such as family and friends. This category explores participants' accounts of the various social supports available to them and their impact; and discusses the various dimensions of the financial burden imposed as a result of CLBP.

All the participants described some sort of support received from family members and friends. This support was rated by all the participants as a vital part of their CLBP journeys. Family members and friends recommended healthcare services, alternative/local/herbal medicines, advice on possible causes and coping strategies.

A friend asked me to buy this spray medication. (Hisses) so I bought it… Even though people advise me to take herbal, I haven't done it (P9S1).

Importantly, all the participants described how their spouses, children and friends provided psychosocial support in the form of encouragement and support with managing CLBP (eg, using ointments to massage their backs). The family also provided help with Activities of daily living (ADLs), including their mobility.

So, do this daddy don't do this. So, they (family) are of support. It really helps. I can't quantify it. But if you ask me on about a scale of 10, I'd put it at 7. Yes, even somebody tells you you've got to go for your medicals today it's something (P1S2).

In a bid to show care, the family sometimes reinforced FABs and passive strategies. This was particularly evident in the narratives of elderly participants.

The children asked me not to be lifting too much heavy things, I shouldn't be washing, cooking … if nobody is in the house, I try to do a little (P9S1).

Spousal support differed across genders, as another form of support was provided by wives. Three male participants reported how their wives had to leave their jobs to be with them to provide physical and psychological support, after they have been advised to avoid activities.

My wife had to stop her work and come and be with me. I'm active but … I was advised not to carry heavy things. Sometimes, you know with this sickness you go through stress, so you need somebody to be with you. Psychologically to reduce the stress (P13S2).

The majority of the participants acknowledged a consequence of CLBP was increased financial burden. FABs related to avoidance/modification of work roles resulted in loss of income or reduction in income. Family and friends therefore provided financial support to help offset healthcare costs and support dependent children/ families.

…so, it was later someone suggested helping me. Coming for physiotherapy, taking the X-ray and all of that, it's that person who bears the cost (P11S1).

Another dimension of financial burden related to the need to employ the services of others to provide help with work and home activities (eg, farming, driving, sweeping and washing) that participants could no longer perform.

Because of the pain I'm experiencing I have to employ someone to come and sweep for me and do my cleaning for me (P1S1).

The financial burden imposed by CLBP appeared to be worsened by participants' biomedical beliefs and acquired biomedical beliefs/FABs from HCPs that led to avoidance of work, repeated imaging, medications, and dependence on the healthcare system. A main source of financial burden was 'health shopping' facilitated by participants themselves, family/friends and HCPs. 'Health shopping' is seeking for healthcare or a remedy from various sources (Bunzli *et al*).[15] In this study, participants' health shopping resulted in costs incurred from laboratory tests, X-ray and MRI, herbal and orthodox medication, and transportation.

… I used to report to the hospital occasionally, then I'd go for an X-ray. The doctor will give me medications. But… I wasn't seeing any improvement. My friend told me about a scientific herbal clinic. At the herbal clinic they carried out a lot of tests on me and I bought medications (P2S2).

## DISCUSSION

The psychosocial impact of CLBP recorded in this study reflected most of the findings recorded from systematic reviews on the impact/experiences of CLBP, although these have mainly involved studies from developed countries.[13–15] The encompassing nature of pain, affecting all spheres of life physical, social and psychological reported by the majority of participants in this study was similarly recorded in a range of studies conducted in developed and low-income and middle-income countries (eg, Brazil, Spain, Canada and Nigeria).[13–15 26 34 52] The current study highlights participants' feeling of loss of self and roles—a loss of their identities. This is highlighted in other studies as a 'feeling of disconnectedness'[53]; unable to perform social roles/ culturally defined roles[25 54] and an oppressive intrusion on the self.[14] Although identity is a multifaceted construct, studies on chronic illness and identity indicate that patients regard work and relationships as important constructs of identity.[55 56] Loss of self and roles, which was a major impact of CLBP described by the current study participants, partly originated from biomedical/biomechanical beliefs acquired from HCPs. Therefore, it appears that the practice of integrating patients back to work, or advice to return to work/activity as early as possible, as suggested in guidelines and studies conducted in developed countries[17] is deficient within Ghana. Early return to work mitigates the effects of the feeling of loss, and enhances function and psychosocial outcomes.[57] Facilitation of early return to work is particularly important for a low-to-middle-income country such as Ghana where there is an existing economic burden and the majority of individuals (89% of the total workforce) rely on informal employment (eg, farming/trading) as a means of sustenance.[58] Informal employment represents employment that is not covered/insufficiently covered by pension, medical insurance, with no entitlement to paid annual leave or sick leave.[58] Furthermore, the study conducted by Charmaz[59] portends that people's interactions within the sociocultural environment preserves the self. These social interactions include work, hobbies, relationships and group memberships. Therefore, strategies that consistently sever or reduce these social interactions potentially undermine the self and deepen the loss of self.

Feelings of negative emotions (such as anger) that were recorded in this study have similarly been recorded in studies on the lived experiences of CLBP conducted in developed countries.[34 60–62] The current study findings and findings from the Nigerian study highlighting participants' extreme negative thoughts around CLBP suggest substantial negative psychosocial impact of CLBP in African countries. Some participants in the current study relayed their fears concerning disability and the use of aids. This fear of disability and possibility of using aids may be linked to Ghanaian cultural representations that associate disability with supernatural causes and stigmatisation has been reported in a critical review of physical disability, rights and stigma in Ghana.[63] However, other studies conducted in developed countries have also reported patients' fear of future disability.[14 25 64 65] Furthermore, lack of support from work and the financial impact of CLBP were reported by the current study participants

and studies conducted in developed countries[53] and Nigeria.[34] Lack of support from work increased feelings of distress, loss and stigma in the current study and other studies.[53] Social support is considered to moderate LBP psychological outcomes, particularly depression.[66] There is also strong evidence that occupational-related psychosocial factors, such as job dissatisfaction and lack of social support affect LBP outcomes.[67] Participants in the current study, however, expressed a reliance on family and friends for sustenance, with no formal support systems in place, especially for self-employed/informally employed individuals. This demonstrates the potential for increased negative psychosocial impact of CLBP in low-to-middle-income countries, such as Ghana.

An important source of loss of income highlighted by participants in the current study was the loss of working time facilitated by numerous/multiple physiotherapy sessions/hospital visits. This, together with the reported influence of HCPs' on FABs, and feelings of loss, distress, stigmatisation and fear, situates HCPs as important mediators of the psychosocial course of CLBP in the Ghanaian context. Other studies have similarly reported HCPs' contributions to patients' biomedical/biomechanical perspectives and the negative impact of this on patient's psychosocial states and CLBP outcomes.[15 68] The current study findings reinforce the notion of CLBP as an 'iatrogenic disorder',[15 24] considering the reported substantial contributions of HCPs (biomedical/biomechanical inclinations) to the psychosocial impact of CLBP among patients. Current management strategies and treatment guidelines for the management of CLBP in developed countries suggest the incorporation of self-management and a biopsychosocial approach to care for patients presenting with CLBP, due to the complex nature of chronic pain and the multidimensional effects of CLBP.[12 17 69]

The current study identified family/friends as a positive sociocultural aspect of the pain experience/coping. The provision of motivation, listening and understanding by family and friends was similarly reported by Bailly et al,[25] in their study on the impact of CLBP. De Souza and Frank[70] in their qualitative study on the impact of CLBP on family and employment also reported spousal support as an important source of support valued by participants. However, in the current study, an extended impact of CLBP on wives' livelihoods/careers was mentioned, as some male participants reported that their wives had to suspend their careers to provide spousal support. This highlights sociocultural aspects of the CLBP experience within Ghana and the far-reaching negative consequences of CLBP on the patient and their family. Lin et al[24] also highlighted a gendered impact of CLBP, with wives altering their lives to take care of their husbands and CLBP impacting on different activities for men and women. The impact of CLBP on domestic chores was a challenge mentioned by all women in the current study, while a few men mentioned the impact of CLBP on their supportive role with domestic tasks. The gendered impact of CLBP has been similarly reported in

studies on lived experiences of CLBP among Punjabis living in the UK[48] and Iranian women.[71]

Although all the current study participants expressed feelings of loss, expressions of guilt/shame about familial support were absent in their narratives, highlighting aspects of disparities between this study and previously conducted studies in developed countries.[13] Interestingly, this study reported how family and friends sometimes facilitated avoidance of activities (passive coping), in a bid to show care. This was in contrast with other studies conducted in developed countries where family and friends reportedly encouraged performance of activities (active coping), and this was embraced by patients as a part of LBP management and a form of distraction from pain.[14 25] This demonstrates disparities in LBP coping strategies/understandings among Ghanaians and other developed countries; and depicts the impact of sociocultural influence on the experience of LBP. Additionally, Rodrigues-de-Souza[52] indicated that the familial environment introduced negative psychosocial aspects of CLBP (eg, feeling misunderstood, which led to withdrawal and accusations of malingering). Reports of feeling misunderstood were reported by a few participants in the current study within the family space. The findings of stigmatisation/marginalisation reported in the current study were similarly depicted in an Australian study,[72] Canadian study,[26] Ethiopian study[35] and systematic reviews on the impact/experiences of CLBP.[13–15 27] An aspect of stigmatisation reported in the current study arose from sociocultural dispositions that situated CLBP as a condition that may not be a serious illness. Some previous studies have indicated the negative effects of overmedicalisation of CLBP, as this is thought to facilitate overdependence on the healthcare system, overdiagnosis/overtreatment and avoidance of activities.[73–75] However, the current study indicates the tendencies for beliefs that serve to delegitimise patients pain to contribute to adverse psychosocial consequences. Indeed, studies have indicated that CLBP patients want their pain to be legitimised through diagnosis and acceptance from important others (eg, family/work).[13 53 76] Acknowledgement of and being empathetic towards patients' suffering as well as providing patient education that explains the complex nature of CLBP of is an important aspect of CLBP management.[20 77] However, legitimisation of CLBP through the emphasis on biomedical findings (as predominantly expressed in the current study) contradicts the current models for assessment and management of CLBP and facilitates adverse outcomes such as increased pain, disability and adverse psychosocial consequences. The current study also recorded the impact of sociocultural beliefs on LBP related fear. Sociocultural beliefs/practices (such as beliefs around the usefulness of herbal medication; fear of toxicity resulting from taking prescribed medications) as recorded in this study may affect healthcare seeking behaviours and engagement with other forms of therapy.

## Strengths and Limitations of the Study
This study was conducted using rigorous qualitative methods, which enhances the reliability of the study

findings. This study demonstrates that similar adverse psychosocial consequences of CLBP pertains among individuals in developed countries and an African country, while highlighting important disparities that exist. The varied sampling strategies (purposive, maximum variation, theoretical) and multiple geographical locations used in this study enriched the breadth and depth of the findings derived in this study and thus the reliability of the study. This study used induction which ensured that the findings reflected participants' own voices. However, the findings may not be applicable to Ghanaian patients with CLBP who have not accessed the Ghanaian healthcare system or patients with CLBP in other African countries, due to contextual and structural differences. Conducting member-checking through returning of findings to participants could have improved the confirmability and thus the validity of this study.

## CONCLUSION

This study depicts the multidimensional (psychosocial) levels of affectation of CLBP (domestic tasks, employment, family/gender/cultural obligations, self, emotional consequences, financial consequences, stigmatisation/marginalisation) in persons living with CLBP in an African country, and the factors underlying these (patients' and HCPs' biomedical/biomechanical beliefs, FABs, sociocultural beliefs, socioeconomic circumstances). This study demonstrates the need for HCPs (physiotherapists and doctors) to consider the multidimensional nature of the experience of CLBP and the need for assessment of psychosocial factors, as part of routine CLBP management. HCPs need to reconsider the biomedical/biomechanical underpinnings driving the psychosocial course of CLBP and coping mechanisms (eg, avoidance of activities) used by patients with CLBP in Ghana. This study reinforces the need for incorporation of a holistic model of care—a biopsychosocial model of care in the management of CLBP in Ghana. This may potentially enhance patient outcomes and address the negative consequences of inherited biomedical/biomechanical beliefs and approach to care highlighted by the study participants. Stigmatisation, marginalisation and narratives of passive coping highlighted by the current study participants suggest the need for population level education on CLBP to improve distorted beliefs around CLBP and its management, as highlighted in this study. Furthermore, there is the need for formal support systems for individuals experiencing CLBP to help mitigate the negative psychosocial impact and burden associated with LBP.

**Acknowledgements** We acknowledge the participants and gatekeepers at both hospitals for their contributions towards this research. We also acknowledge Margaret Asare and Emmanuel Baidoo for their help during back translation.

**Contributors** JAA, FM and CD conceived and designed the study. JAA collected the data and is the guarantor of the study. JAA and PKA were involved with validation and the back translation of transcripts and topic guides. JAA was responsible for transcription, leading data analysis and initial drafting of the article. All authors contributed to analysis, interpretation and manuscript development. All authors approved the final submitted manuscript.

**Funding** This publication was funded by London South Bank University.

**Competing interests** None declared.

**Patient and public involvement** Patients and/or the public were involved in the design, or conduct, or reporting, or dissemination plans of this research. Refer to the Methods section for further details.

**Patient consent for publication** Consent obtained directly from patient(s).

**Ethics approval** This study involves human participants and the study protocol was approved by the University of Nottingham Faculty of Medicine and Health Sciences ethics committee (REF: 931808), and the research and development, and institutional research boards of both study sites (IRB/000136/2018). Participants gave informed consent to participate in the study before taking part.

**Provenance and peer review** Not commissioned; externally peer reviewed.

**Data availability statement** No data are available. No additional data available.

**ORCID iD**
Josephine Ahenkorah Ampiah http://orcid.org/0000-0003-1752-2027

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
