## [Reviewer comments · BMJ Open]

ARTICLE DETAILS

TITLE (PROVISIONAL)	A Qualitative Investigation of the Psychosocial Impact of Chronic Low Back Pain in Ghana
AUTHORS	Ampiah, Josephine; Moffatt, Fiona; Diver, Claire; Ampiah, Paapa Kwesi

VERSION 1 – REVIEW

REVIEWER	Ikemoto, Tatsunori Aichi Medical University
REVIEW RETURNED	07-Apr-2023

GENERAL COMMENTS	This study investigates the psychosocial impact of CLBP among patients with CLBP in Ghana. Although this study appears to contain novelty, it does not adequately describe the problems behind it or the methods used. Some revisions are required to guarantee its quality. Abstract What is HCPs? What is this an abbreviation for? Introduction As the authors described, a plenty of studies reported psychosocial problems among patients with CLBP, particularly in developed countries. I think the novelty of this study is its impact in African countries. Two articles have already reported psychosocial impacts of CLBP among patients with CLBP in Nigeria(28) and Ethiopia(29), however, cultural differences between these two African countries and Ghana is not clearly described at all. First, please describe the characteristics of back pain patients in Ghana, describe common back pain treatments, and describe cultural characteristics. For example, what are the differences in these contexts compared to developed countries. Method & Results The detailed data analysis procedures are not adequately presented. For example, an appendix should present the actual content of how many interviews were conducted and what words were recovered from Subject 1. In this context, categorical words should be underlined and categorized. Ex, Category1, Loss of self and roles. Authors described that "All the participants described some sort of loss resulting...", however, only several examples were mentioned in the text. Please present selection of sentences that fall into Category 1 in an appendix Table 1 for all subjects. Please summarize each table by category, with an appendix showing which subjects made which statements.
---

	Discussion P9, Line13-14 : " partly originated from acquired fear-avoidance beliefs (FABs) " There is no evidence to indicate this in the current study.
--	---

REVIEWER	Petrucci , Giorgia University of Rome, Department of Orthopaedic and Trauma Surgery
REVIEW RETURNED	18-Apr-2023

GENERAL COMMENTS	Dear researchers, Thank you for the opportunity to review your manuscript entitled “A Qualitative Investigation of the Psychosocial Impact of Chronic Low Back Pain in Ghana”. It is an interesting and well written review that has some details to be checked before it can be published:  • Page 4, line 22: You should correct the term “focussed” • Page 4, line 34-36: You should better explain the aim of the study in the introduction. • Page 4, line 40-41: you should deepen the mean of “broader research”, giving more details about this study. • Page 5, line 11-12: You should rewrite the period using another verb, for example “the study was set in two Ghanaian hospital” not “were used”. • Page 13, line 35: You should cite Charmaz as “the study conducted by Charmaz”. • Page 21: the sum of the percentage of different ages is 90%, please correct numbers with the right descriptive data. Probably the error is in the age 60-69. All the best in your submission!
---

VERSION 1 – AUTHOR RESPONSE

Reviewer 1’s comments	Authors’ response
This study investigates the psychosocial impact of CLBP among patients with CLBP in Ghana. Although this study appears to contain novelty, it does not adequately describe the problems behind it or the methods used. Some revisions are required to guarantee its quality.	Suggested revisions have been made throughout the manuscript.
Abstract What is HCPs? What is this an abbreviation for?	Full meaning of abbreviation has been provided in the abstract. Page 2, line 48
Introduction As the authors described, a plenty of studies reported psychosocial problems among patients with CLBP, particularly in developed counties. I think the novelty of this study is its impact in African countries. Two articles have already reported psychosocial impacts of CLBP among patients with CLBP in Nigeria (28) and Ethiopia(29), however, cultural differences between these two African countries and Ghana is not clearly described at all. First, please describe the characteristics of back pain patients in Ghana, describe common back pain treatments, and describe cultural characteristics. For example, what are the	Characteristics of back pain patients in Ghana and Africa have been described. Page 3, lines 75-86 The common treatment approaches used in Ghana and Africa have been described and contrasted with developed countries. Page 3, lines 94-104 The subjectivity of chronic pain experience and the differences in cultural and structural experiences in different African countries have been highlighted. Page 3, lines 132-144

differences in these contexts compared to developed countries.	
Method & Results The detailed data analysis procedures are not adequately presented. For example, an appendix should present the actual content of how many interviews were conducted and what words were recovered from Subject 1. In this context, categorical words should be underlined and categorized. Ex, Category1, Loss of self and roles. Authors described that "All the participants described some sort of loss resulting...", however, only several examples were mentioned in the text. Please present selection of sentences that fall into Category 1 in an appendix Table 1 for all subjects. Please summarize each table by category, with an appendix showing which subjects made which statements.	A table (supplemental file 1) containing derived categories, concepts, codes, mechanisms, and exemplar quotes has been provided. This gives a clear trail of how information was extracted from participants' narrations and how the categories were derived from the participants' data. A statement has been added to clarify category 1. Page 7, lines 278-279
Discussion P9, Line13-14 : " partly originated from acquired fear-avoidance beliefs (FABs) " There is no evidence to indicate this in the current study.	This has been removed.
Reviewer 2's Comments	Authors' response
Page 4, line 22: You should correct the term "focussed"	"Focussed" has been replaced with "mostly reported" Page 3, line 123
Page 4, line34-36: You should better explain the aim of the study in the introduction.	This has been done. Page 3, lines 144-146
Page 4, line 40-41: you should deepen the mean of "broader research", giving more details about this study.	This has been clarified. Page 4, lines 148-150
Page 5, line 11-12: You should rewrite the period using another verb, for example "the study was set in two Ghanaian hospital" not "were used".	Another phrase has been used. Page 5, line 174
Page 13, line 35: You should cite Charmaz as "the study conducted by Charmaz".	This has been done. Page 13, line 512
Page 21: the sum of the percentage of different ages is 90%, please correct numbers with the right descriptive data. Probably the error is in the age 60-69.	The percentage for the 60-69 has been corrected. Page 22